# Effect of Multi-Strain Probiotic Supplementation on URTI Symptoms and Cytokine Production by Monocytes after a Marathon Race: A Randomized, Double-Blind, Placebo Study

**DOI:** 10.3390/nu13051478

**Published:** 2021-04-27

**Authors:** Edgar Tavares-Silva, Aline Venticinque Caris, Samile Amorin Santos, Graziela Rosa Ravacci, Ronaldo Vagner Thomatieli-Santos

**Affiliations:** 1Department of Bioscience, Universidade Federal de São Paulo, Santos 11015-020, Brazil; tavares.silva@unifesp.br (E.T.-S.); samile.amorin@gmail.com (S.A.S.); 2Department of Psychobiology, Universidade Federal de São Paulo, São Paulo 04032-020, Brazil; nutri.alineventicinque@gmail.com; 3Department of Gastroenterology, Medicine Faculty, University of São Paulo, Santos 11015-020, Brazil; grazielametanutri@gmail.com

**Keywords:** nutrition, probiotics, immunology, strenuous exercise, monocytes, URTI, marathon race

## Abstract

(1) Purpose: Performing strenuous exercises negatively impacts the immune and gastrointestinal systems. These alterations cause transient immunodepression, increasing the risk of minor infections, especially in the upper respiratory tract. Recent studies have shown that supplementation of probiotics confers benefits to athletes. Therefore, the objective of the current study was to verify the effects of probiotic supplementation on cytokine production by monocytes and infections in the upper respiratory tract after an acute strenuous exercise. (2) Methods: Fourteen healthy male marathon runners received either 5 billion colony forming units (CFU) of a multi-strain probiotic, consisting of 1 billion CFU of each of *Lactobacillus acidophilus* LB-G80, *Lactobacillus paracasei* LPc-G110, *Lactococcus* subp. *lactis* LLL-G25, *Bifidobacterium animalis* subp. *lactis* BL-G101, and *Bifidobacterium bifidum* BB-G90, or a placebo for 30 days before a marathon. Plasma cytokines, salivary parameters, glucose, and glutamine were measured at baseline, 24 h before, immediately after, and 1 h after the race. Subjects self-reported upper respiratory tract infection (URTI) using the Wisconsin Upper Respiratory Symptom Survey (WURSS—21). The statistical analyses comprised the general linear model (GLM) test followed by the Tukey post hoc and Student’s *t*-test with *p* < 0.05. (3) Results: URTI symptoms were significantly lower in the probiotic group compared to placebo. The IL-2 and IL-4 plasma cytokines were lower 24 h before exercise, while the other cytokines showed no significant differences. A lower level of IL-6 produced by monocytes was verified immediately after the race and higher IL-10 at 1 h post. No differences were observed in salivary parameters. Conclusion: Despite the low number of marathoners participating in the study, probiotic supplementation suggests its capability to preserve the functionality of monocytes and mitigate the incidence of URTI.

## 1. Background

The physiological changes caused by a marathon can generate adverse effects in the gastrointestinal tracts of athletes, ranging from abdominal pain to diarrhea and vomiting [1]. This activity increases intestinal permeability [2] and endotoxemia, featuring a higher plasma concentration of lipopolysaccharides (LPS) [3,4].

In addition, the marathon generates a type of immune depression called ‘open window’ after exercise [5,6]. A transient immunodepression characterizes this period, thus creating favorable conditions for viruses and bacterial infections after the race, especially in the upper respiratory tract [7,8,9].

Monocytes circulate in the bloodstream, acting quickly and with a nonspecific response [10]. After being stimulated by LPS, monocytes quickly produce a lot of inflammatory cytokines, such as IL-1 alpha and beta, IL-6, IL-8, IL-12, and TNF alpha; and anti-inflammatory cytokines, such as IL-10 and TGF-β [11,12]. The inflammatory cytokines are produced early after exposure to an antigen, while the production of anti-inflammatory cytokines to control and inhibit inflammation, mainly IL-10, occurs later [11,12].

After the marathon race, monocytes increase the release of lysosomal enzymes and reactive oxygen species (ROS), accompanied by cytokine production to degrade and remove cellular debris and damaged tissue. An increase in *Monocyte Chemotactic Protein 1*—MCP1—is also observed, known as an activator and chemotactic factor of monocytes, promoting activation and cell extravasation to the injured tissues [13]. However, strenuous physical exercise results in transient immunodepression [6] and the consequent appearance of minor infections, notably in the upper respiratory tract [14].

Nutritional strategies can mitigate the stress in the immune system generated by exercise, such as macro and micronutrient supplementation. Carbohydrate and glutamine supplementation modulate cellular activity and the endocrine system [15]. Glutamine is a nonessential amino acid with the highest concentration in skeletal muscle and plasma, being essential for the proliferation of immune cells [16]. Carbohydrates strongly influence energy [17,18] and cellular metabolism [15]. These strategies are intended to reduce the risk of minor infections, especially in the upper respiratory tract [19].

Supplementation with probiotics has demonstrated benefits for the immune system and gastrointestinal system [19]. These benefits include improved intestinal functionality [20] and an increase in circulating leukocytes [21,22,23]. Probiotics also stimulate the maturation of immune cells present in the intestinal epithelium, such as lymphocytes, macrophages, and dendritic cells [24,25]. Despite additional evidence on the benefits of probiotics in health, the effectiveness of probiotic supplementation depends on different factors, such as strains, dose, and different components following the production of bacteria [26].

Some studies investigated the effects of probiotic supplementation on the immune system cells, notably in athletes after strenuous physical exercise [27]. Moreira et al. [28] showed that three months of supplementation with two milk-based fruit drinks with 3 × 10^8^ CFU per day of *Lactobacillus GG (LGG)* (ATCC53103) had no effects compared to a placebo on the prevalence of asthma and allergy during the pollen season. Kekkonen et al. [29] also evaluated supplementation of 4 × 10^10^ CFU per day of *Lactobacillus GG (LGG)* on upper respiratory tract infections and gastrointestinal symptoms in 141 athletes for three months before the race and two weeks after. There were no differences between placebo and probiotics; however, the authors found shorter GI symptom duration. The work developed by O’Brien et al. [30] demonstrates that fermented milk Kefir, containing the microbial culture of *lactic acid bacteria* 10^10^ CFU, consumed for 15 weeks, was capable of improving performance in a 1.5 mile time trial and attenuated the inflammatory levels generated by intensive exercise.

Supplementation of a multi-strain capsule containing 10 × 10^9^ CFU *Lactobacillus acidophilus* CUL-60 NCIMB 30157, 10 × 10^9^ CFU *Lactobacillus acidophilus* CUL-21 NCIMB 30156, 9.5 × 10^9^
*Bifidobacterium bifidum* CUL-20 NCIMB 30172, 0.5 × 10^9^
*Bifidobacterium animalis* subsp *lactis* CUL-34 NCIMB 30153, and 55.8 mg.d-1 fructooligosaccharides appears not to result in a higher concentration of eHsp72 in the bloodstream after a marathon race [31]. On the other hand, supplementation of a multi-strain probiotic with two doses per day of 1.66 × 10^9^ CFU *Bifidobacterium bifidum* W23, 1.66 × 10^9^ CFU *Bifidobacterium lactis* W51, 1.66 × 10^9^ CFU *Enterococcus faecium* W54, 1.66 × 10^9^ CFU *Lactobacillus acidophilus* W22, 1.66 × 10^9^ CFU *Lactobacillus brevis* W63, and 1.66 × 10^9^ CFU *Lactococcus lactis* W58 for 14 weeks generated benefits for intestinal permeability and, possibly, for inflammation [32].

Other studies observed positive effects on immune response and gastrointestinal functionality. Two studies evaluated the effects of probiotic supplementation with *Lactobacillus fermentum* VRI-003 (PCC) 1.26 × 10^10^ CFU for four weeks and 1 × 10^9^ CFU *Lactobacillus fermentum* VRI-003 (PCC) for 11 weeks in athletes. The authors observed a lower incidence of minor infections in the upper respiratory tract [33,34]. These results were similar to supplementation with 6.5 × 10^9^ CFU *Lactobacillus casei Shirota* [35] and *Lactobacillus helveticus Lafti L10* 2 × 10^10^ CFU/day for 14 weeks [36].

Recently, Zhang et al. [37] demonstrated that supplementation with 3 × 10^7^
*CFU Lactobacillus paracasei*, 3 × 10^7^ CFU *Lactobacillus casei* 431^®^, and 3 × 10^6^ CFU *Lactobacillus fermentium* PCC reduces the risk of URTI and flu-like symptoms and increases the concentration of sIgA in the gut. Pugh et al. [38] showed that probiotic supplementation with 25 × 10^9^
*Lactobacillus acidophilus* CUL60, *L. acidophilus* CUL21, *Bifidobacterium bifidum* CUL20, and *Bifidobacterium animalis* subsp. *lactis* CUL34 can reduce gastrointestinal symptoms and severity in marathon runners, as described in Table 1. However, none of these works demonstrate any parallel with the cellular production of cytokines.

In this context, there is little evidence about the effects of probiotic supplementation on cytokine production by monocytes after a marathon race. Therefore, the hypothesis of the present study considered that probiotic supplementation would attenuate the immunodepression generated after a marathon race, decreasing the incidence of minor infections and preserving monocyte functionality.

## 2. Methods

### 2.1. Participants

This study was approved by the Research Ethics Committee of UNIFESP/Hospital São Paulo (REC #691.519—2014) and complied with the Brazilian legislation’s norms in resolution *N*. 466/12 of the National Health Council. The participants signed the term of free and clarified consent (FCC) in agreement with the study, whose flow is reported in the diagram of Figure 1. The sample was composed of marathon runners, males, who performed aerobic training at least three times a week. The mean age of the groups was 39.92 ± 3.47 years, with a body mass of 74.84 ± 7.23 kg, height of 177.5 ± 4.49 cm, BMI of 23.99 ± 1.99 (Bod Pod^®^, Life Measurement Inc., Concord, CA, USA), and Vo2peak of 55.72 ± 7.46 kg/mL/min. The average marathon race time was 247.9 ± 35.80 min, with an average speed of 10.57 ± 1.45 km/h. The groups were randomized and double-blinded for supplementation.

Individuals were excluded from the study if they presented health problems after medical evaluation, anti-inflammatory drugs, antibiotics, smoking, licit and illicit drug use, or use of nutritional supplements, except carbohydrates, during the experiment period (Figure 1).

### 2.2. Experimental Design

The participants performed cardiovascular tests (resting electrocardiogram, exertion, and ergospirometry) and the VO_2_ peak test and received the supplementation. After thirty days of supplementation, they ran a marathon—the International Marathon of São Paulo, 2015 (42.1 km). The initial temperature during the test was 21.5 °C and air humidity was 67%. The marathon was a time trial protocol, and during the race, carbohydrate with a concentration of 6% was offered at different distances (km 18, 29, 31, and 36). The division of the groups was randomized, and the supplementation was double-blind, as described below (Figure 2).

### 2.3. Supplementation

In the probiotic group, for 30 days, seven volunteers consumed a supplement of the multi-strain probiotic in gelatinous capsules consisting of 1 billion CFU *Lactobacillus acidophilus-LB-G80*, 1 billion CFU *Lactobacillus paracasei-LPc-G110*, 1 billion CFU *Lactococcus* subp. *lactis-LLL-G25*, 1 billion CFU *Bifidobacterium animalis* subp. *lactis-BL-G101*, and 1 billion CFU *Bifidobacterium bifidum-BB-G90*, totaling at the beginning of the supplementation 5 × 10^9^ CFU 2.0 g/day per day, ingested during the nighttime period in the same way as the placebo. The probiotic capsules were prescribed by a nutritionist and made in a Compounding Pharmacy from CEDRONI, São Paulo/SP—Brazil. Before manufacture by the Compounding Pharmacy, all the bacteria strains were analyzed, reported, approved, and certificated according to the level of genus/species based on 16 s rRNA sequencing by LEMMA SUPPLY SOLUTIONS LTDA—Jardinópolis/SP—Brazil.

The control group consisted of seven volunteers who consumed the placebo supplementation for thirty days (2.0 *g* of corn starch per day), in gelatinous capsules, similar in color, taste, smell, and size to the capsules consumed by the probiotic group, ingested during the nighttime period.

### 2.4. Determination of VO_2_Peak

A progressive test was performed on a treadmill, after a warm-up at an initial velocity of 7.0 km/h for three minutes and with a load increment of 1.0 km/h per minute until voluntary exhaustion. A fixed slope of 1% was used throughout the test. The tests were performed in a laboratory with standardized acclimatization (T = 22 °C and RH = 61%). Respiratory parameters were collected and analyzed by a gas analyzer COSMED^®^ model Quark PFT—Pulmonary Function Testing—FRC & DLCO—Rome, Italy. A Hans Rudolph^®^ flow-by face mask was used. All calibration procedures followed the manufacturer’s recommendations.

### 2.5. Blood Collection

Blood samples were collected four times: before supplementation (baseline), one day before the marathon (24 h before), post-race (immediately after), and 1 h after the marathon (1 h post). At each of the four moments, we collected 30 mL of blood from the median cubital vein (10 mL of the blood was used for the cellular assays, and 20 mL of remaining blood was centrifuged at 690× *g* for 15 min at 4 °C). Plasma or serum was extracted in Hemogard EDTA K2 Plus 4 mL (7.2 mg of EDTA K2—DB, Curitiba, PR—Brazil) and DB SST II Advance—DB, Curitiba, PR—Brazil tubes (Serum separator tube) 5.0 mL PLUS, aliquoted and stored in a freezer at −80 °C for subsequent dosages.

### 2.6. Monocyte Isolation

The separation of cells occurred from 10 mL of the total blood in tubes with 3 mL of Histopack 1077 and 3 mL of Histopack 1119 after 30 min of centrifugation at 400× *g*. The layer corresponding to leukocytes was transferred to sterile conical tubes and diluted in PBS. The tubes were centrifuged again at a speed of 400× *g* at 4 °C for 10 min. Subsequently, a 1 h incubation was performed on monocytes adhered to the plaque. The supernatant was composed of 2 mL of culture medium RPMI-1640 (enriched with 1 mL of serum from the volunteer and 650 µL of penicillin) and 500 mL cell suspension leukocytes in the wells. The total number of cells used in the culture assay was 1 × 10^6^ in each well. After cell separation, monocytes were stimulated with 0.2 mL of LPS 5μg/mL and incubated for a period of 24 h, in a greenhouse, at 37 °C, at a concentration of 5.0% CO_2_. All supernatants were aliquoted in Eppendorf tubes of 2000 µL and frozen in the freezer at −80 °C until analysis. Brand reagents Sigma-Aldrich (Merck group), San Luis, MI, USA.

### 2.7. Cytokine Production by Monocytes Stimulated by LPS

**Cytokine production:** After 24 h incubation, the concentrations of IL-1beta, IL-6, IL-10, and TNF-α cytokines in the supernatant were evaluated by the Multiplex assay with Millipore kits, Darmstadt, Germany. Simultaneous analysis of multiple cytokines was performed with Luminex technology with magnetic beads. The Luminex 200™ Analyzer was used with the MAGPIX^®^ system using the software MILLIPLEX^®^ Analyst 5.1.

### 2.8. Determination of Serum and Plasmatic Parameters

**Glucose:** The glucose concentration was determined enzymatically with Bioclin kits (glucose monoreagent). Belo Horizonte—MG-Brazil.

**Glutamine:** The plasma glutamine concentration was determined by the enzymatic method according to the method described by Windmueller and Spaeth [39] and adapted from Cooney, Davis and Van Atta [40].

**Cytokines:** IL-2, IL-4, IL-10, and TNF-α concentrations were evaluated by the Multiplex assay with Millipore kits, Darmstadt, Germany. Simultaneous analysis of multiple cytokines was performed with Luminex technology with magnetic beads. The Luminex 200™ Analyzer was used with the MAGPIX^®^ system using the software MILLIPLEX^®^ Analyst 5.1.

### 2.9. Determination of Salivary Parameters

The saliva samples were collected using the Salivet method (cylindrical roller bearings that absorb saliva over the period of one minute) for four moments: thirty days before the exercise (baseline), one day before the exercise (24 h before), post-exercise (immediately after), and one hour after the exercise (1 h post). After collection, the sample was placed into a tube and centrifuged at a speed of 600× *g* for 20 min. A transparent fluid specimen was obtained and stored frozen (−80 °C) for analysis.

**IgA**: IgA was determined by an immunoturbidimetric method using kits from Labtest^®^ (Lagoa Santa, Brazil). IgA flow rates were calculated by multiplying the concentration of each parameter by salivary flow (mL/min), as described by Usui et al. [41].

### 2.10. Food Record

For dietary data during the supplementation period, a questionnaire was completed twice a week and once on weekends. This dietary record characterized the volunteer’s dietary profile in the supplementation period, and the analysis was performed through the Dietotherapy^®^ system by a nutritionist.

### 2.11. Upper Respiratory Tract Infection Questionnaire (URTIq)

Monitoring of the symptoms of upper respiratory tract infections was performed for seven days after the marathon using the Wisconsin Upper Respiratory Symptom Survey Questionnaire—21 (WURSS—21), proposed by Barret et al. [42] and translated for the Brazilian Portuguese population by Moreira and Cavazzoni [43]. The questionnaire contained 21 questions of the survey type, seeking to evaluate health-related information through common cold symptoms. Nine symptoms were evaluated (blocked nose, sneezing, sore throat, scratchy throat, cough, hoarseness, head congestion, chest congestion, and feeling tired) with seven different severity options (very mild to severe). All items in this questionnaire are based on a Likert-type scale from 0 to 7.

### 2.12. Statistical Analysis

The data are expressed as mean ± standard deviation. The statistical tests were applied according to the normality of the data, verified by the Shapiro–Wilk test. We used the analysis of repeated measures GLM with Tukey post hoc. The Student’s *t*-test was used to perform the statistical analysis of common cold symptoms and severity, subject characteristics, food record, and body composition. The significance level adopted was *p* ≤ 5%. The SPSS 21 software was used to perform the analyses.

## 3. Results

In the present work, marathon runners were supplemented for thirty days and performed the International Marathon Race of São Paulo. Runners who participated in the analysis till the end were homogeneous and included seven volunteers per research arm. The descriptive analysis of the sample is presented in Table 2. The ergospirometry data are shown in Table 3. In the food consumption analysis, no significant differences were found between the groups regarding the ingestion of carbohydrates, proteins, and lipids and the total calorie value, as shown in Table 4.

The production of cytokines was evaluated after the stimulus by LPS. A significant increase in the production of IL-1beta was observed in the probiotic group 1 h post compared to 24 h before (*p* < 0.02) and immediately after the race (*p* < 0.03), as shown in Figure 3A. Regarding the production of IL-6, there was a difference between the groups at baseline (*p* < 0.005) and immediately after (*p* < 0.03). There was a significant decrease in the placebo group at the moment 24 h before compared to baseline (*p* < 0.007) and an increase immediately after (*p* < 0.003) and 1 h post (*p* < 0.03), compared to 24 h before (Figure 3B). In the production of IL-10, significant differences were observed between groups at 1 h post (*p* < 0.05). A significant increase in IL-10 occurred in the probiotic group immediately after and 1 h post compared to baseline (*p* < 0.04 and *p* < 0.01) and 24 h before (*p* < 0.01 and *p* < 0.002) (Figure 3C). No differences were observed in TNF-alpha production between groups and moments (Figure 3D).

Regarding the plasma cytokine concentrations, a significant decrease in IL-2 was observed in the probiotic group at 24 h before compared to baseline (*p* < 0.04). There was a difference between the groups at 24 h before (*p* < 0.04) (Figure 4A). Concerning IL-4, differences were observed between groups at 24 h before (*p* < 0.02). A significant decrease was also observed in the placebo group at the time 24 h before compared to baseline (*p* < 0.04). Furthermore, there was a significant decrease 1 h post compared to immediately after in the same group (*p* < 0.02) (Figure 4B).

The plasma concentration of IL-10 in the probiotic group significantly decreased after the supplementation period (*p* < 0.001). However, IL-10 increased immediately after (*p* < 0.001) and 1 h post (*p* < 0.007). In the placebo group, there was a significant increase immediately after compared to 24 h before (*p* < 0.008) (Figure 4C). In both groups, TNF-alpha increased immediately after compared to 24 h before (placebo *p* < 0.02; probiotic *p* < 0.02). In the placebo group, the 1 h post value was significantly different from the 24 h before value (*p* < 0.02) (Figure 4D).

When evaluating the energetic substrates in the probiotic group, there were no significant differences in the plasma concentration of glutamine. However, in the placebo group, we observed a significant decrease immediately after (*p* < 0.01) compared to 24 h before, and this decrease lasted until 1 h post (*p* < 0.04) (Figure 5A). Plasma glucose concentration did not change between the different times and groups (Figure 5B).

After the marathon race, the volunteers responded for seven consecutive days to the Wisconsin Upper Respiratory Symptom Survey—21 (WURSS—21) to assess the incidence and severity of symptoms. There was a lower incidence in the probiotic group (29%), as shown in Figure 6A. All volunteers (100%) of the placebo group presented at least one symptom during the seven days of evaluation, while in the probiotic group, approximately 71% presented at least one symptom.

Regarding the number of symptoms that both groups presented, we found that in the first two days (day 1 *p* < 0.01; day 2 *p* < 0.02), the probiotic group presented significantly more symptoms than the placebo group. However, the number of symptoms decreased on the third day, although not significantly, compared to the placebo. On the fifth (*p* < 0.001) and sixth days (*p* < 0.02), the probiotic group did not present any further symptoms, while in the placebo group, the symptoms increased (Figure 6B). Considering the severity of these symptoms, it was verified that in the probiotic group, the symptoms presented during the seven days were less severe on the third (*p* < 0.04), fifth (*p* < 0.001), sixth (*p* < 0.001), and seventh days (*p* < 0.03) when compared to the placebo group (Figure 6C).

When assessing the mucosal immunity (Figure 7), no significant differences between the groups were observed regarding IgA concentration. However, in the placebo group, the IgA concentration immediately after was significantly lower than 24 h before (*p* < 0.001), but the modification was reestablished 1 h post. In the probiotic group, a significant decrease was observed only at 1 h post when compared to 24 h before (*p* < 0.02) and immediately after (*p* < 0.01), as shown in Figure 7A. The salivary flow in the placebo group decreased 24 h before compared to baseline (*p* < 0.001); for both groups, there was a significant decrease immediately after (placebo *p* < 0.007; probiotic *p* < 0.001) and 1 h post (placebo *p* < 0.04; probiotic *p* < 0.004) compared to 24 h before, as shown in Figure 7B. The secretion ratio in the placebo group decreased 24 h before compared to baseline (*p* < 0.005); for both groups, there was a significant decrease immediately after (placebo *p* < 0.002; probiotic *p* < 0.001) and 1 h post (placebo *p* < 0.01; probiotic *p* < 0.001) compared to 24 h before, as shown in Figure 7C.

## 4. Discussion

The results showed that supplementation of multi-strain probiotics consisting of 1 billion CFU *Lactobacillus acidophilus-LB-G80*, 1 billion CFU *Lactobacillus paracasei-LPc-G110*, 1 billion CFU *Lactococcus* subp. *lactis-LLL-G25*, 1 billion CFU *Bifidobacterium animalis* subp. *lactis-BL-G101*, and 1 billion CFU *Bifidobacterium bifidum-BB-G90* totaling 5 × 10^9^ CFU 2.0 g/day for thirty days was able to modulate the production of cytokines by monocytes at different times and significantly decrease the incidence and severity of symptoms in the upper respiratory tracts of athletes exposed to a marathon race.

In the first two days, the probiotic group presented a higher number of symptoms. However, no differences were observed between the groups on the third day and there was a reduction on the fifth day until no more symptoms were reported in the supplemented group. In contrast, in the placebo group, the number of symptoms increased. Moreover, on all days after the marathon, the symptoms in the supplemented group were less severe than in the placebo group. The improvement concerning URTI symptoms was independent of IgA salivary concentration, considering that no significant differences were detected between the groups.

The marathon race is an exhaustive exercise, decreasing the immune response and increasing susceptibility to minor infections in the upper respiratory tract [6]. Our protocol confirms that strenuous exercises, such as a marathon, can generate transient immunodepression, verified by the symptoms and severity in the days after the race.

In both groups, we verified the presence of symptoms of minor infections. However, the incidence of URTI symptoms in the probiotic group was 29% lower when compared to placebo, and during the seven days, we observed a lower number of symptoms and less severity in the supplemented group. These findings corroborate previous studies demonstrating that athletes supplemented with probiotics have less minor infections [34,44].

Although some studies use supplementation for prolonged periods [36,45]. Our study is in agreement with previous research that also used thirty days of supplementation to verify the benefits of probiotics, with an interaction of the gastrointestinal tract and immune system [33,44]. It has been suggested that a minimum period of thirty days is necessary for the effective colonization of bacteria in the intestine. The results found in this supplementation period are significant, considering the complexity of the probiotic–organism interactions, price of supplementation, and possibly greater adherence by the athletes to interventions with probiotics.

The human microbiome is formed by different commensal, symbiotic, pathogenic bacteria, fungi, and viruses, living in a ‘mini-ecosystem’, and disturbances can significantly impair the individual [25]. This ecosystem is considered an organ with different functions, conferring benefits to the host through digestion, immune system stimulation, and prevention against pathogens through the intestinal epithelium [25,46,47]. Modulation of the intestinal microbiota occurs through dietary adjustments that include modification of food composition as well as the inclusion of probiotics [48]. The intake of digestible and non-digestible carbohydrates combined with probiotic supplementation may contribute to intestinal health [49]. In our study, both groups presented higher carbohydrate intake, demonstrating food equity and highlighting the possible influence of probiotic supplementation on the presented results.

Probiotics act mainly in the gastrointestinal tract by different mechanisms that include the interaction and modulation of the intestinal microbiota and production of metabolites such as short-chain fatty acids—SCAFs (butyrate, acetate, and propionate). In addition, probiotics increase the bioavailability of nutrients and assist the homeostatic development of the resident microbial environment by stimulating immunological functions that include cell maturation, proliferation, and IL-10 production [49,50]. Intestinal health is relevant for maintaining the body and immunological homeostasis [51]. The possible interaction of probiotics on the intestinal microbiota could justify our results of a lower incidence and severity of URTI symptoms in the supplemented group, confirming the previous studies [33,34,36,37,44].

Because of the benefits that the microbiome can confer on the organism [52], we believe that our results of URTI symptoms were not exclusively due to probiotic supplementation, but also due to monocytes and plasma cytokines that mediate cell functionality. Cytokines are cell messengers with a high regulatory capacity for inhibition and stimulation of the immune response [53].

Clancy et al. [22] verified alterations in the immunological response through the production of IFN-g by lymphocytes after probiotic supplementation *(Lactobacillus acidophilus* LAFTI^®^L10 2 × 10^10^ CFU/day). Similarly, Cox et al. [33] also reported increased secretion of IFN-g and decreased URTI of athletes after supplementation for one month of *Lactobacillus fermentum* VRI-003 1.2 × 10^10^ CFU/day. Haywood et al. [44] showed that multi-strain supplementation with *Lactobacillus gasseri* (2.6 × 10^9^ CFU/day), *Bifidobacterium bifidum* (0.2 × 10^9^ CFU/day), and *Bifidobacterium longum* (0.2 × 10^9^ CFU/day) for thirty days was effective to attenuate gastrointestinal and respiratory symptoms, as well as the duration of the symptoms, in athletes.

In our study, after exposure to LPS, monocytes responded differently in the groups. This difference was verified through the production of cytokines in the cell supernatant. After exposure to LPS, cytokines organize the immune response through autocrine, paracrine, and endocrine mechanisms [54]. The lipopolysaccharides comprise part of the extracellular membrane of Gram-negative bacteria recognized by cells of the innate immune system, mainly by monocytes, triggering the production of different cytokines and functionalities, such as phagocytosis [55,56].

We observed significant differences between the groups regarding the production of IL-6 immediately after and of IL-10 at 1 h post. Lower IL-6 values and higher IL-10 values were verified in the probiotic group compared to the placebo. In addition, we observed a significant increase in IL-1 in the probiotic group at the time of 1 h post compared to other moments. Although no studies have evaluated the production of cytokines by monocytes after supplementation with probiotics, it is known that monocytes generally present different characteristics according to the cellular environment in which they are inserted [10].

IL-6 is considered a regulatory cytokine, acting on the acute phase of inflammation and orchestrating the anti-inflammatory process [57]. Rossol et al. [12] describe that viable monocyte production after exposure to LPS occurs at different times. Higher concentrations of inflammatory proteins are observed in the first hours, while IL-10, an anti-inflammatory cytokine, increases after approximately 20 h. Another explanation for our results on IL-6 and IL-10 concentrations between the groups may be correlated to the different monocyte profiles identified as classic monocytes (CD14hiCD16-) and non-classic monocytes (CD14lowCD16 + or CD14hiCD16 +) [58,59]. Thus, we believe that the presence of IL-10 indicates a final resolution of the cellular challenge after exposure to LPS in the probiotic group. On the other hand, the higher concentration of IL-6 in the placebo group indicates an initial final phase of the exposure of the monocytes to LPS considering that no significant increase in inflammatory cytokines was observed.

After performing a strenuous exercise, the toll-like receptors (TLR), especially those of type 4 (TLR-4), in monocytes are less expressed, which may consequently decrease the sensitivity of these cells to respond to the stimulus of LPS and produce smaller amounts of pro-inflammatory cytokines [58]. Thus, we believe that the monocytes of the probiotic group have preserved functionality compared to the placebo. The increase in IL-1 and IL-10 production after the marathon race reflects the cytokine production balance after LPS exposure [12]. These differences found in cytokine profiles between groups could represent a slower response of monocytes exposed to LPS, impacting the immunosurveillance of the placebo group and possibly affecting the common cold symptoms and severity after the marathon race.

During physical exercise, there is also synthesis and release of cytokines by different tissues that reflect the plasma concentration and may influence the cellular responses [60,61]. Previous studies corroborate our findings, indicating alterations in cytokines after performing strenuous exercises, increasing IL-10 and TNF-alpha [61,62,63].

Other studies also verified changes in plasma concentrations of IL-2 and IL-4 after physical exercise [61,64,65,66]; however, we did not observe these alterations in our study. We found significant differences between the groups only at 24 h before regarding the concentrations of IL-2 and IL-4. These modifications 24 h before in both groups demonstrate a change in the immune profile. After thirty days of probiotic supplementation, the plasma concentration of IL-2 decreased. This result could indicate a systemic anti-inflammatory effect [67]. The IL-2 is mainly produced by Th1 lymphocytes acting concomitantly with other inflammatory cytokines such as INF-gamma and TNF-alpha. Interestingly, there is a balance between IL-2 and IL-4. Therefore, when IL-2 has a higher concentration, the concentration of IL-4 decreases [60]. However, we did not find this increase in IL-4 concentration.

In addition to these factors, we observed that probiotic supplementation led to maintenance of the plasma concentration of glutamine at the different moments analyzed. In contrast, in the placebo group, the concentration was lower immediately after and 1 h post compared to baseline and 24 h before. The plasma glutamine concentration can directly impact the immunological cells [68,69]. High amounts of glutamine are used for cellular proliferation and cytokine production [16,70,71]. An in vitro study conducted by Murphy and Newsholme [72], demonstrated that macrophages use high glutamine rates during phagocytosis activities, and production of cytokines and reactive oxygen species, and that this use is even higher after exposure to LPS.

Prolonged exercise is associated with a decrease in these amino acid plasma levels, which is one of the possible causes of immunosuppression generated by strenuous physical exercise, increasing the susceptibility to minor infections [16,73]. Our results corroborate the literature, considering a lower incidence of symptoms and severity in the supplemented group, concomitant with maintenance of the plasma glutamine concentration. Simultaneously, in the placebo group, there was a lower concentration of this amino acid after the marathon race and higher susceptibility to minor infections. This indicates that strenuous and long-lasting exercises decrease the plasma concentration of glutamine, affecting the immune response [69,73].

In addition, glutamine is important to the intestinal epithelium as it is an energetic supply for the cells that comprise this tissue, such as secretory, immunologic, neuroendocrine, and enterocyte cells [74]. There was no significant reduction in glutamine in the probiotic group. Supposedly, the intestinal microbiota of this group was more balanced and did not require as much glutamine from the bloodstream to maintain the homeostasis of the region. The maintenance of glutamine concentration suggests that the gastrointestinal tract did not suffer much from the impacts of the marathon race.

Concerning the plasma concentrations of glucose over the different moments, we did not find differences between them. Maintenance of the glucose concentration possibly occurs through carbohydrate consumption during the marathon race (concentration of 6%—18, 29, 31, and 36 km). This result equalizes the groups with respect to the energy metabolism of glucose for both physiological and cellular purposes. These results corroborate the scientific literature indicating an increase in or maintenance of circulating glucose levels in supplemented athletes [75,76].

Considering all the factors mentioned above, we could also explain the URTI symptom results through the worsening of mucosal immunity, notably by the association between URTI and IgA concentration [14]. One of the mechanisms of mucosal protection is by the rate of IgA [66], in addition to a strong relationship between the concentration of IgA and salivary flow with the risk of minor infections [77,78,79,80]. However, our results do not confirm this relationship between the symptoms presented and the salivary parameters analyzed because we did not verify significant differences between the groups. Our salivary results confirm the work published by Cox et al. [33] who observed a decrease in URTI symptoms in the group supplemented with probiotics, without differences in concentrations of IgA compared to the placebo group. Thus, we suggest that the immunomodulatory effect of probiotic supplementation is independent of the oral mucosa immunity.

## 5. Conclusions

Considering the data obtained through different tests, and despite the low number of cases analyzed, we suggest that probiotic supplementation could effectively attenuate the symptoms and severity of minor infections in the upper respiratory tract of marathon athletes by preserving immune response and cell functionality. There were no alterations in the salivary parameters between the groups. Probiotic supplementation is an efficient, easy to apply, and effective strategy to mitigate the deleterious effects caused by strenuous exercises on the immune system.

## 6. Study Limitations

This work did not evaluate the intestinal microbiota of the athletes. Therefore, we cannot verify the modifications/alterations in the intestinal epithelium and the composition of intestinal microbiota. According to the institution’s ethical protocol, in which the participants had the possibility to withdraw without prejudice, and as described in the Free and Clarified Consent, only 14 volunteers (placebo = 7 and probiotic = 7) out of 46 adhered to the study until the end.

## Figures and Tables

**Figure 1 nutrients-13-01478-f001:**
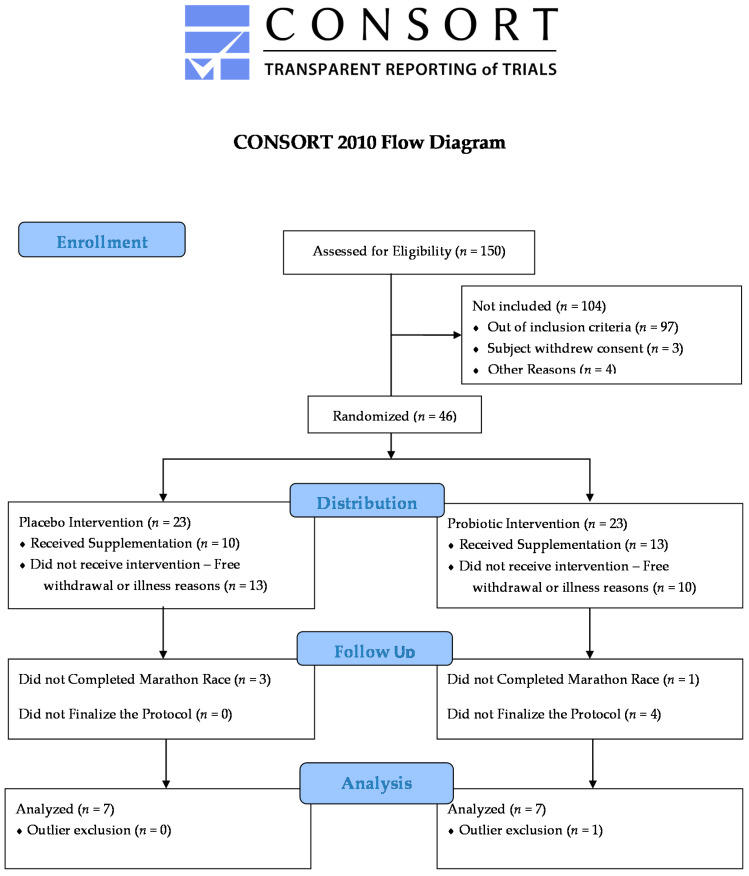
Flow diagram.

**Figure 2 nutrients-13-01478-f002:**
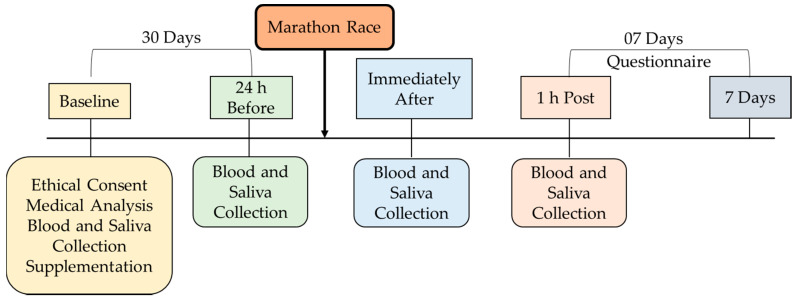
Experimental design.

**Figure 3 nutrients-13-01478-f003:**
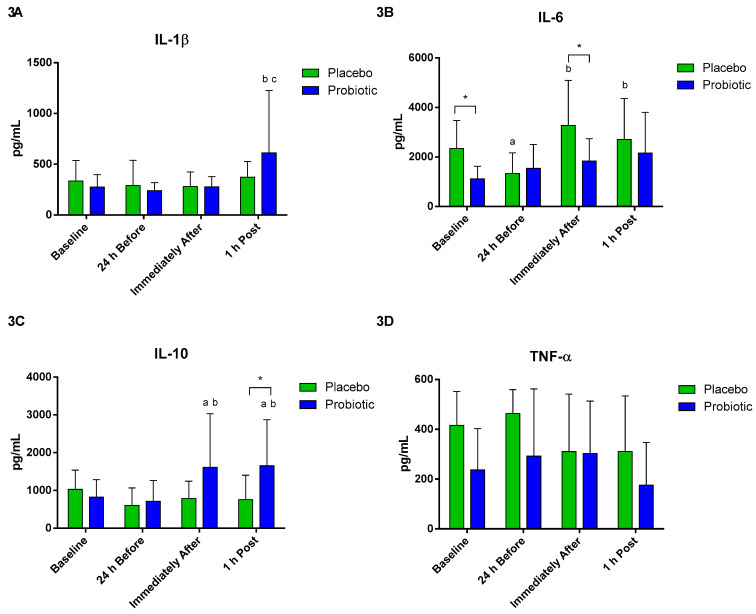
Cytokine production by monocytes. Data presented as mean ± standard deviation. Placebo (*n* = 7) and probiotic (*n* = 7). The comparison between groups and moments was performed through the general linear model (GLM) with post hoc Tukey. (**A**). IL-1β production. (**B**). IL-6 production. (**C**). IL-10 production. (**D**). TNF-α production. a: differs from the ‘Baseline’ of the same group. b: differs from the ‘24 h Before’ of the same group. c: differs from the ‘Immediately After’ of the same group. * difference between groups.

**Figure 4 nutrients-13-01478-f004:**
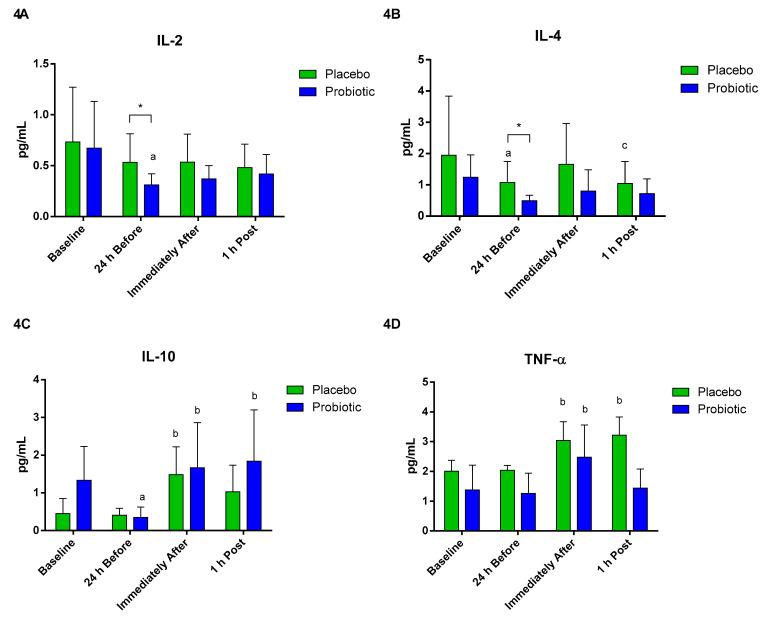
Plasma cytokine concentration. Data presented as mean ± standard deviation. Placebo (*n* = 7) and probiotic (*n* = 6). The comparison between groups and moments was performed through the general linear model (GLM) with post hoc Tukey. (**A**) IL-2 concentration. (**B**) IL-4 concentration. (**C**) IL-10 concentration. (**D**) TNF-α concentration. a: differs from the ‘Baseline’ of the same group. b: differs from the ‘24 h Before’ of the same group. c: differs from the ‘Immediately After’ of the same group. * difference between groups.

**Figure 5 nutrients-13-01478-f005:**
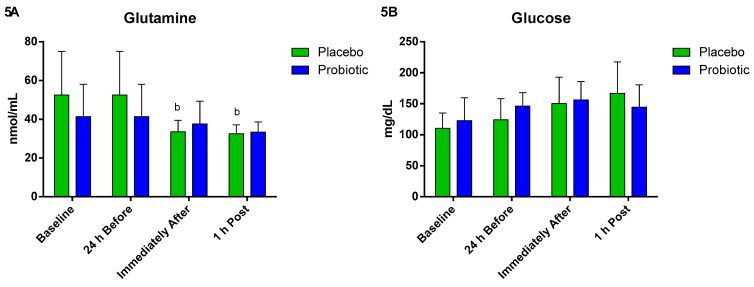
Plasma glutamine and glucose concentration. Data presented as mean ± standard deviation. Placebo (*n* = 7) and probiotic (*n* = 7). The comparison between groups and moments was performed through the general linear model (GLM) with post hoc Tukey. (**A**) Glutamine concentration. (**B**) Glucose concentration. b: differs from the ‘24 h Before’ of the same group.

**Figure 6 nutrients-13-01478-f006:**
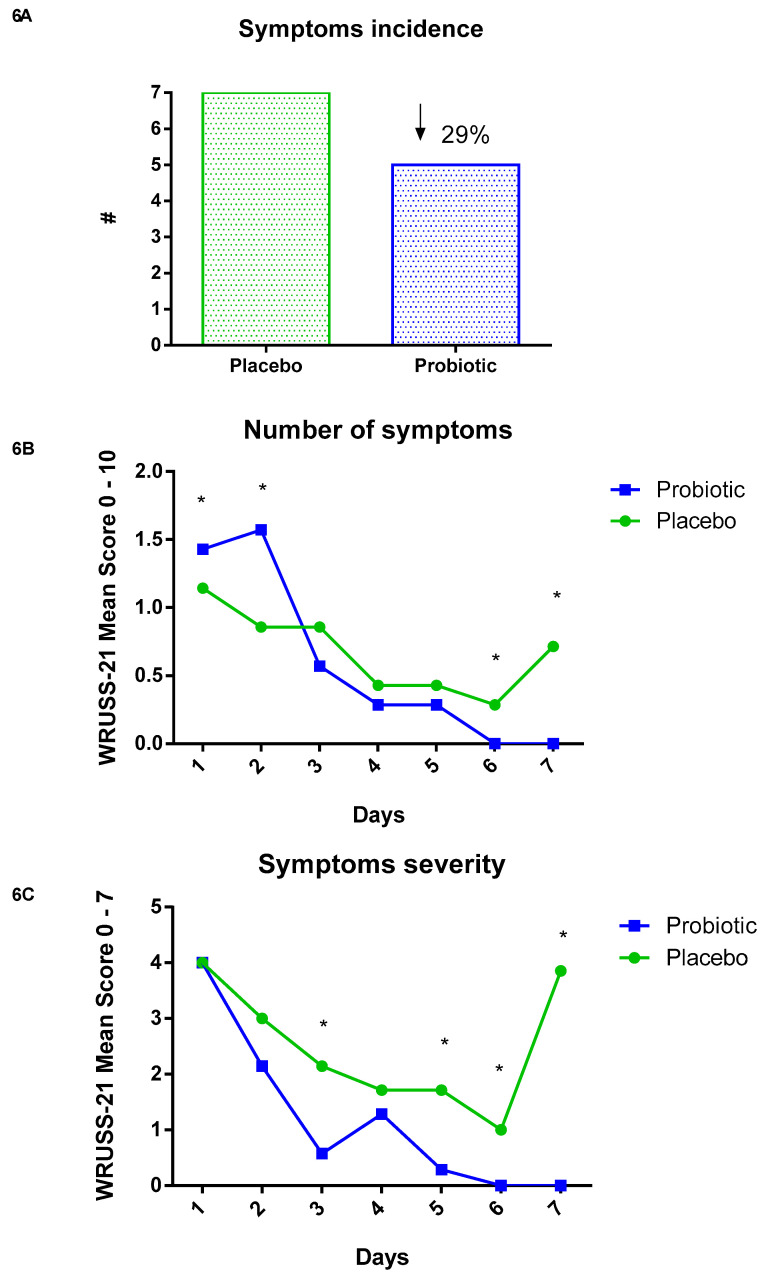
Wisconsin Upper Respiratory Symptom Survey (WURSS—21). Data presented as # and mean of symptoms and severity. Placebo (*n* = 7) and probiotic (*n* = 7). The comparison between groups was performed through the Student *t*-test. (**A**) Symptoms incidence. (**B**) Number of symptoms. (**C**) Symptoms severity. * Difference between groups.

**Figure 7 nutrients-13-01478-f007:**
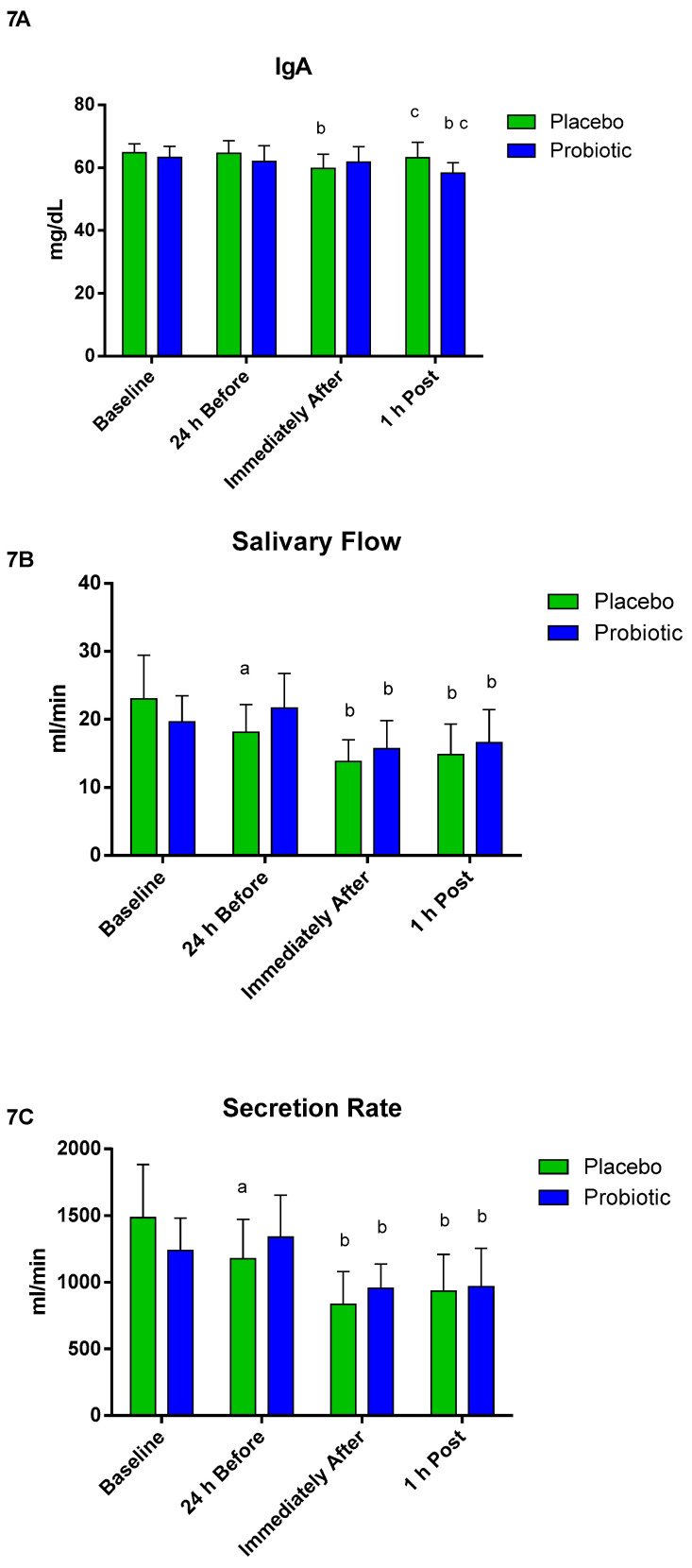
Salivary parameters. Data presented as mean ± standard deviation. Placebo (*n* = 7) and probiotic (*n* = 7). The comparison between groups and moments was performed through the general linear model (GLM) with post hoc Tukey. (**A**) IgA concentration. (**B**) Salivary flow. (**C**) Secretion rate. a: differs from ‘Baseline’ of the same group. b: differs from the ‘24 h Before’ of the same group. c: differs from the ‘Immediately After’ of the same group.

**Table 1 nutrients-13-01478-t001:** Summary table of probiotics and effectiveness on symptoms of URTI.

Authors	Probiotic	Dose	Time	Results
Moreira et al., 2007	*L. GG (LGG)*	3 × 10^8^ CFU	12 weeks	↔ prevention allergic markers
Kekkonen et al., 2007	*L. GG (LGG)*	4 × 10^10^ CFU	12 weeks	↔ symptoms respiratory infections
Cox et al., 2010	*L. fermentum VRI-003*	1.26 × 10^10^ CFU	16 weeks	↓ incidence ↓ severity URTI
West et al., 2011	*L. fermentum VRI-003*	1 × 10^9^ CFU	11 weeks	↓ incidence GI symptoms ↓ incidence URTI
Gleeson et al., 2011	*L. casei shirota LcS*	6.5 × 10^9^ CFU	16 weeks	↓ frequency URTI
Lamprecht et al., 2012	*B. bifidum W23* *B. lactis W51* *E. faecium W54* *L.acidophilus W22* *L. brevis W63* *L. lactis W58*	1 × 10^10^ CFU	14 weeks	Benefits on intestinal permeability
West et al., 2014	*B. animalis* subsp. *lactis Bl-04*	2 × 10^9^ CFU	23 weeks	↓ risk URTI
Haywood et al., 2014	*L. gasseri* *B. bifidum* *B. longum*	2.6 × 10^9^ CFU 0.2 × 10^9^ CFU 0.2 × 10^9^ CFU	4 weeks	↓ incidence ↔ severity URTI
Michalickova et al., 2016	*L. helveticus lafit L10*	2 × 10^10^ CFU	14 weeks	↓ duration ↓ symptoms URTI
Zhang et al., 2018	*L. paracasei* *L. casei 431* *L. fermentium PCC*	3 × 10^7^ CFU 3 × 10^7^ CFU 3 × 10^6^ CFU	12 weeks	↓ symptoms ↓ risk URTI

Caption: ↓ Reduce. ↔ No effect.

**Table 2 nutrients-13-01478-t002:** Subjects’ characteristics.

Variable	Placebo *n* = 7	Probiotic *n* = 7	*p*-Value
Age (years)	38.28 ± 3.09	41.57 ± 3.20	0.075
Race Time (min)	243.0 ± 33.73	252.87 ± 39.77	0.62
Average Speed (km/h)	10.73 ± 1.53	10.41 ± 1.48	0.70
Body Mass (kg)	78.43 ± 8.40	71.24 ± 3.55	0.059
Height (cm)	179.36 ± 5.23	175.82 ± 3.01	0.14
BMI (kg/m²)	24.90 ± 1.81	23.08 ± 1.83	0.087
Fat Mass (%)	19.7 ± 6.87	14.48 ± 3.63	0.10
Free Fat Mass (%)	80.3 ± 6.87	85.5 ± 3.63	0.10
Fat Mass (kg)	16.03 ± 6.29	10.95 ± 2.29	0.068
Free Fat Mass (kg)	64.47 ± 8.49	60.77 ± 4.27	0.32

Data presented as mean ± standard deviation. The comparison between groups was performed using the Student’s *t*-test. Caption: BMI—body mass index.

**Table 3 nutrients-13-01478-t003:** Ergospirometry data.

Variable	Placebo *n* = 7	Probiotic *n* = 7	*p*-Value
VO_2Peak_ (kg/mL/min)	54.53 ± 6.88	56.92 ± 8.35	0.57
VO_2Peak_ (L/min)	4.16 ± 0.45	4.08 ± 0.70	0.79
Maximum Speed (Km/h)	17.16 ± 1.57	17.57 ± 1.27	0.60
Maximum HR (Bpm)	182.16 ± 10.05	178.7 ± 3.45	0.40
VE Maximum (L/min)	143.05 ± 21.01	139.68 ± 10.52	0.71
LT1 VO_2_ (kg/mL/min)	41.73 ± 7.58	42.80 ± 9.25	0.81
LT2 VO_2_ (kg/mL/min)	48.88 ± 6.87	49.37 ± 8.49	0.90
LT1 HR (Bpm)	154.83 ± 10.44	151.71 ± 5.76	0.50
LT2 HR (Bpm)	170.0 ± 7.52	163.85 ± 5.89	0.11

Data presented as mean ± standard deviation. The comparison between groups was performed using the Student’s *t*-test. Caption: VO_2_—oxygen consumption. HR—heart rate. VE—ventilation. LT—ventilatory threshold.

**Table 4 nutrients-13-01478-t004:** Food record.

	Placebo *n* = 7	Probiotic *n* = 7	*p*-Value
	**Kcal**	**Kcal**	
TCV (kcal)	1994.46 ± 365.73	2434.69 ± 505.53	0.087
	%	%	
Carbohydrate	47.77 ± 4.27	47.88 ± 16.98	0.98
Proteins	18.92 ± 1.62	17.53 ± 4.07	0.42
Lipids	33.28 ± 2.76	34.56 ± 12.96	0.80
	**Grams**	**Grams**	
Carbohydrate	237.46 ± 61.11	294.52 ± 122.76	0.29
Proteins	92.66 ± 7.10	105.95 ± 38.37	0.38
Lipids	74.88 ± 10.65	92.52 ± 51.14	0.38

Data presented as mean ± standard deviation. The comparison between groups was performed using the Student’s *t*-test. Caption: TCV—total calorie value.

## Data Availability

The datasets used and/or analyzed during the current study are available from the corresponding author on reasonable request.

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
