# Peer review of "Effect of Multi-Strain Probiotic Supplementation on URTI Symptoms and Cytokine Production by Monocytes after a Marathon Race: A Randomized, Double-Blind, Placebo Study"

_nutrients, 2021, doi:10.3390/nu13051478_

Round 1

Reviewer 1 Report

The paper is well written, please pay attention to some typing errors, hopefully.
The author is asked to provide a summary table of the microbiota strains useful and not useful for preventing URTI.
In discussion How do the authors explain what the influence of diet can be on the modifications of the microbial and its interactions with the modifications induced by probiotes introduced by integration? 

In discussion how do you explain the role of IL2 and TNF?

Reviewer 2 Report

Strenuous exercise impacts on the function of the immune and gastro-intestinal systems, by increasing the risk of minor infections, especially of the upper respiratory tract.

In this article, the authors try to verify the influence of a 30 days probiotic blend supplementation on the production of cytokines by monocytes of 14 males who underwent a marathon and on URTI

In my opinion the study group is too small to draw conclusions.

Moreover, the two sub-groups (placebo vs probiotics supplemented sub-group) present different cytokines production by monocytes and plasma cytokines concentrations before probiotics supplementation (baseline, Figure 3). How do the Authors explain this?

The body text and the abstract are rather confused, mainly in the results and discussion/conclusion paragraphs. I suggest the Authors to completely rephrase them.

Moreover, there are a lot of inaccurancies. I would encourage the Authors to review the English language and the grammar used. As examples..Authors often use genitive improperly. There are a number of sentences in which wrong tenses or singular terms instead of plural and viceversa are used. Some phrases are rather confused and too long.

Authors should always use italics forms for bacteria genus and species and write in full the first time they mention them

Authors should avoid to use unknown abbreviations in the abstract.

Sometimes the Authors write in the first person plural, sometimes in the third person

For homogeneity, in line 147, Authors should write the distance of the marathon in km

After mention each reagent or disposable used, Authors should always write the name of the Company, City and State

Line 197 0.2 mL instead of 0,2

Line 245 p value is incorrect

In all p values after the zero Authors should use a point, not a comma

Reviewer 3 Report

To estimate the benefits of probiotic supplementation on severe physical stress, the authors studied the effects of multi-strain probiotic supplementation on marathon runners. The results indicated that the intake of probiotic bacteria consisting of 5 species for 30 days before a marathon suppressed symptoms in the upper respiratory tract after the marathon. I agree with the conclusion of the study (Fig. 6), but I could not understand the mechanism and logic of the study as follows.

  • The amount of cytokine produced by stimulating monocytes separated from volunteers in vitro was evaluated (Fig. 3). I could not understand the results well because the description of the experimental method is insufficient (Line 187-199). Are the amounts of cytokines shown in Fig. 3 secreted from the same number of monocytes or from monocytes prepared from the same volume of blood. There is no data about the number of monocytes in the assay or in the peripheral blood.
  • LPS was chosen for monocyte stimulation. Does different forms of stimulation; for example, bacterial DNA, induce different responses in monocytes?
  • Please provide a brief overview of the WURSS-21 in Methods. Is there any difference in the kinds of symptoms observed between the probiotic and placebo groups?

The score for the placebo group was elevated at day 7 (Fig.6). Are these symptoms induced by infection? We need detailed information on the types and transition of symptoms. Without this information, we cannot discuss which function was stimulated by the intake of probiotics and therefore plays a role in the suppression of symptoms. The authors should discuss these points to provide a better understanding of the relationship between the respiratory symptoms and immunity as modulated by probiotics.

Round 2

Reviewer 2 Report

Please rephrase the Abstract (line 32) into:

“Despite the low number of marathoners participating in the study, probiotic supplementation suggests its capability to preserve the functionality of monocytes and mitigate the incidence of URTI”

In the Methods section (line 129), please write “The participants signed the term of free and clarified consent (FCC) in agreement with the study, whose flow is reported in the diagram of Figure 1".

In the Results section (line 252) please rephrase into "Runners who joined the analysis till the end were homogeneous and included seven volunteers per research arm".

In the Conclusion section (line 499), please rephrase into “Considering the data obtained through different tests, and despite the low number of cases analysed, we suggest that Probiotic supplementation could effectively attenuate the symptoms and severity of minor infections in the Upper Respiratory Tract of marathon athletes by preserving immune response and cell functionality”.

In the Study limitations section (line 510), please rephrase into “According to the institution's ethical protocol, in which the participants had the possibility to withdraw without prejudice, and as described in the Free and Clarified  Consent,  only 14 volunteers (Placebo = 7 and Probiotic = 7) out of  46 adhered to the study until the end”.
